# An Empirical Study on the Resilience of Partial Merging to Model Clone Attacks

Tiantong Wu [1 2]   Yurong Hao [1]   Wei Yang Bryan Lim [1]

## Abstract

Model merging is a promising technique to enhance the capabilities of neural networks (NNs) by integrating multiple downstream fine-tuned models without requiring access to clients' raw data or substantial computation resources. However, conventional model merging typically requires collecting the full set of fine-tuned parameters from multiple clients, which may expose them to model-privacy risks. An emerging approach, known as *partial model merging (PMM)*, mitigates this risk by splitting the model into private and shared parts, where only the shared part is merged while the private part remains local to each client. Despite its stricter parameter fusion, PMM can still achieve competitive performance compared to full-parameter sharing. However, the privacy properties of PMM remain underexplored. In this paper, we propose a novel model clone attack and assess the risk of reconstructing the unshared private part of a partially merged model under eight attack scenarios with varying prior knowledge (i.e., partial training data, model parameters and/or model structure). Our comprehensive experiments reveal that merging NNs without adequate protection is highly vulnerable. Even when only a small fraction of the training data, model parameters, or model structure is exposed, adversaries can still recover significant portions of the private model's performance.

## 1. Introduction

*Model merging* (aka model fusion) (Yang et al., 2024b; Yadav et al., 2023; Xu et al., 2024) integrates multiple downstream fine-tuned neural network (NN) models with diverse capabilities into a single model without retraining or additional fine-tuning. It enables effective reuse, fusion, and transfer of users' knowledge. Hence, users without relevant domain-specific data can mutually benefit from other users' data, without exchanging their raw data. A widely adopted approach of multi-task model merging is the Task Arithmetic method introduced by Ilharco et al. (2023), where multiple vectorised models (i.e., task vectors) are summed to produce a single merged model. This group of approaches requires collecting the complete set of fine-tuned parameters from multiple entities and then merging these parameters to construct a universal merged model. It is known as full model merging (FMM), as depicted in Figure 1a. However, the domain-specific fine-tuned models are increasingly proprietary and closed-source due to the high costs of data collection and training, making the distribution of full parameter sets impractical in many real-world FMM deployments. Moreover, FMM compromises **model privacy**. An adversary can perform *model clone attacks (aka model stealing attack)* by constructing an alternative NN model (i.e., a surrogate model) that closely mimics behaviours of the victim model (Papernot et al., 2017; Orekondy et al., 2019; Roberts et al., 2019), thereby obtaining a local copy that substitutes for the original victim model without incurring additional cost.

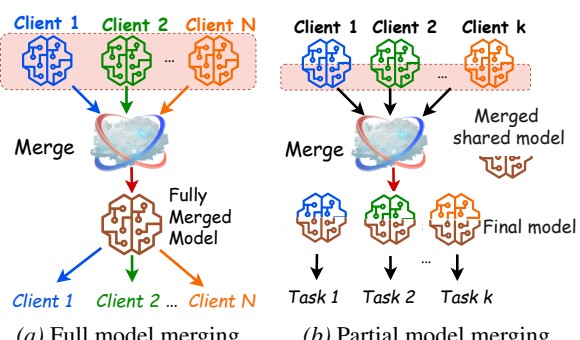

*(a)* Full model merging          *(b)* Partial model merging

*Figure 1.* Full model merging versus Partial model merging. (a) A merged model for $N$ tasks. (b) Assemble $N$ partitioned models with a merged model part (shared part) for $N$ tasks.

*Partial model merging (PMM)* (Stoica et al., 2024; Wang et al., 2025) has emerged as a viable alternative, wherein the full model is partitioned into shared and private parts, as illustrated in Figure 1b. PMM enforces a stricter parameter fusion, merging only the shared parts of the model, while the

[1]College of Computing and Data Science, Nanyang Technological University, Singapore [2]Alibaba-NTU Global e-Sustainability Lab (ANGEL), Nanyang Technological University, Singapore. Correspondence to: Tiantong Wu <tiantong.wu@ntu.edu.sg>.

*Proceedings of the $43^{rd}$ International Conference on Machine Learning*, Seoul, South Korea. PMLR 306, 2026. Copyright 2026 by the author(s).

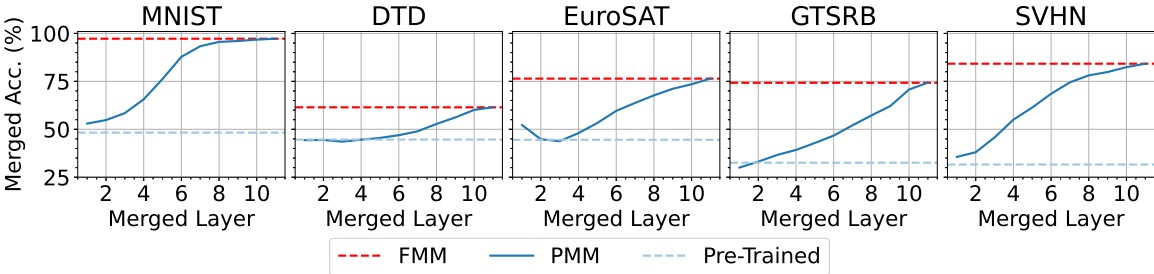

*Figure 2.* Merged ViT-B/32 model accuracy for the partially merged model with different numbers of merged layers. The merging is performed between models fine-tuned on five different downstream tasks (i.e., MNIST, DTD, EuroSAT, GTSRB, and SVHN).

private parts remain with local clients. This design reduces the number of shared model parameters, thereby reducing both overheads and the potential for model-privacy leakage.

Empirically, we observed that PMM can achieve higher model performance than the pre-trained model and closer to, or even surpass, FMM when a larger portion of the model is merged. To illustrate this phenomenon, we use the widely adopted ViT-B/32 model (Radford et al., 2021) and evaluate across five benchmark datasets (i.e., MNIST, DTD, EuroSAT, GTSRB, and SVHN). Specifically, we analyse the accuracy of both partially and fully merged models across multiple tasks by varying the number of merged transformer layers. As shown in Figure 2, the red dashed lines indicate the FMM performance, serving as empirical upper bounds, while the light blue dashed lines indicate the pre-trained model performance before fine-tuning, serving as lower bounds. The solid blue curves trace PMM accuracy for different numbers of merged transformer layers. We observe that the accuracy of the merged model generally increases as more layers are merged across all five datasets. For ViT-B/32, merging 75% of layers from downstream fine-tuned models yields PMM that retains at least 85.89% of FMM's accuracy, while reducing communication and computation costs to about 75% of FMM's costs. Similar trends can be observed for ViT-B/16 and ViT-L/14, with detailed results provided in Appendix A.

PMM induces a distinct partial-sharing setting that creates a meaningful model-stealing attack surface. However, the potential model privacy risks associated with this approach remain unexplored. In particular, PMM changes the volume of model information exposed to an adversary relative to FMM, but it is unclear to what extent sharing parameters incurs model privacy leakage. To the best of our knowledge, no existing work has examined potential model privacy vulnerabilities under PMM, nor the privacy-utility trade-off induced by varying the number of shared parameters. These gaps motivate our central question:

**How would model privacy be affected by sharing a subset of model parameters for merging?**

To make this question concrete, we quantify model privacy risk in terms of how successfully an adversary can extract private model behaviour under PMM. We identify a two-sided information asymmetry: On the adversary side, the victim's training samples, model structure and parameters are largely hidden, which constrains attack design and makes evaluation difficult under realistic assumptions; On the victim side, the adversary's objectives and capabilities are often unknown, which prevents direct measurement of leakage risk and decide the number of layers shared in PMM to balance generalisation and privacy exposure. With this framing, we assess the model-privacy risks of PMM from the adversary's perspective, with access to task-specific components restricted under different knowledge constraints. The detailed contributions of this paper are as follows:

- We conduct the first-of-its-kind systematic privacy analysis of PMM. Our study reveals the inherent vulnerabilities of PMM and demonstrates that an adversary can achieve a model performance comparable to that of the victim's full model.

- We introduce *ModelPirate*, a noval model-clone attack tailored to PMM. The proposed *ModelPirate* aims to recover the behaviour of the private part of the model given limited prior knowledge.

- We evaluate *ModelPirate* across eight attack scenarios with varying degrees of prior knowledge, using diverse model architectures and datasets. Our results offer empirical guidance for attack defence and client layer-sharing decisions.

## 2. Preliminaries

In this section, we formally define PMM. It is commonly known that in an NN model, the layers closer to the outputs contain information more specific to the model's tasks (Nasr et al., 2019; Vandenhende et al., 2022). This insight has recently been applied to the model merging technique to merge only a subset of model parameters or layers (Stoica et al., 2024; Wang et al., 2025). Based on these existing

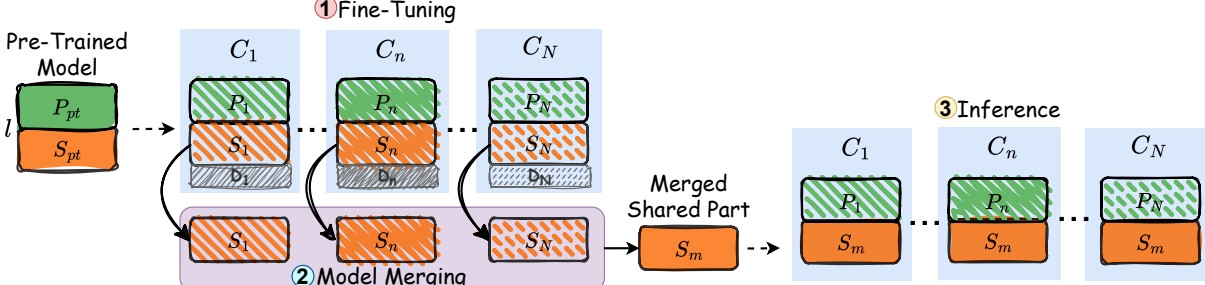

*Figure 3.* Partial model merging. Client $C_n$ fine-tunes the pre-trained model and partitions its fine-tuned model into private part $P_n$ and shared part $S_n$. The merged shared part $S_m$ is connected to the private part $P_n$ at client $C_n$ for inference. Dashed arrows indicate model distributions where the pre-trained/merged model is distributed to all clients $C = \{C_n | n = 1, \cdots, N\}$.

works, we consider PMM clients sharing layers closer to the inputs, while keeping the rest of the layers closer to the outputs private to improve the merged model's generalisation while keeping the task-specific information private. Let a pre-trained base model be

$$f_\theta : \mathcal{X} \to \mathcal{Y}, \tag{1}$$

with $L$ layers and parameters $\theta = (\theta^1, \theta^2, \ldots, \theta^L)$. Each client separates its full individual model into two parts at layer $l$: the *private part* $P(\theta) := \theta^{l+1:L}$ remains private at the client, whereas the *shared part* $S(\theta) := \theta^{1:l}$ is sent to the a merging entity or uploaded to a platform such as Hugging Face [1] for partial open-source. We assume all clients share the same $l$ for merging feasibility.

Let the total number of participating clients be $N$. As illustrated in Figure 3, each client $C_n \in C$ fine-tunes a common pre-trained model $S_{pt} + P_{pt}$ on their own task $T_n$ with data $D_n$, obtaining $\theta_n$ and thus shared part $S_n := S(\theta_n)$, private part $P_n := P(\theta_n)$ (Step ①). Then, the shared parts from all clients $\{S_n\}_{n=1}^N$ are merged as the *merged shared part* [2]

$$S_m = \mathcal{M}(S_1, \ldots, S_N), \tag{2}$$

using the merging algorithm $\mathcal{M}$ (Step ②). Finally, client $C_n$ uses the obtained partially-merged model $S_m + P_n$ for inference (Step ③).

## 3. Our attack: *ModelPirate*

### 3.1. Problem definition

We study a benchmark adversarial setting under PMM with a single adversary-victim pair. As illustrated in Figure 4, client $C_a$ is the *adversary* and client $C_v$ is the *victim*, where

---

[1] https://huggingface.co

[2] In this paper, we consider a general PMM scenario, where the $S_n$ from each client $n$ is sent to a merging entity. The merging entity can be one of the participating clients or a third party (e.g, a cloud server). Then, model merging will be performed at the merging entity after it receives $S_n$ from all clients.

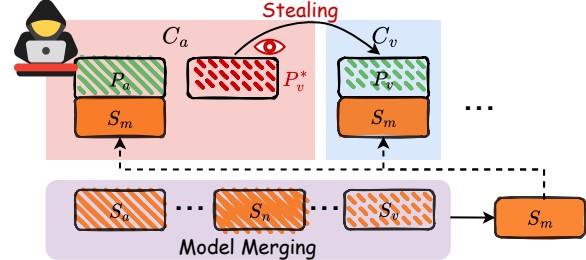

*Figure 4.* Adversarial model. The adversary $C_a$ trains a clone model $P_v^*$ to simulate/steal the behaviour of the victim's $C_v$ private part $P_v$.

$a \neq v$. The two clients are fine-tuned on different downstream tasks, i.e., $T_a \neq T_v$. The adversary is *honest but curious*. It follows the PMM protocol as an ordinary participant while attempting to reconstruct the victim's private model. Following the notation in Section 2, victim $C_v$ holds a fine-tuned model split at layer $l$ into $(S_v, P_v)$. We define the *target model* as

$$f_v(x) := f(x; S_v, P_v). \tag{3}$$

Specifically, we adopt a partially homogeneous PMM setting where all shared parts to be merged $\{S_n\}_{n=1}^N$ are structurally compatible, whereas private parts $\{P_n\}_{n=1}^N$ may be heterogeneous. The adversary's goal is to construct a *clone private model part* $P_v^*$ such that the composed model

$$\tilde{f}_v^\star(x) := f(x; S_v, P_v^*) \tag{4}$$

mimics the behaviour of $f_v(x)$. In particular, for a query distribution $Q$ (e.g., induced by accessible data), we aim for a behavioural discrepancy $\mathbb{E}_{x \sim Q}\left[d\left(f_v(x), \tilde{f}_v^\star(x)\right)\right]$ to be minimal, where $d(\cdot, \cdot)$ denotes a task-appropriate distance.

### 3.2. Threat Model

#### 3.2.1. ADVERSARY'S PRIOR KNOWLEDGE

We distinguish three types of prior knowledge (i.e., *Self-knowledge* $\mathcal{K}_{\text{self}}$, *Shared knowledge* $\mathcal{K}_{\text{shared}}$ and *Auxiliary*

*knowledge* $\mathcal{K}_{\text{aux}}$) available to $C_a$:

- $\mathcal{K}_{\text{self}}$: information inherently possessed by $C_a$, including it's full *source model* $S_a+P_a$, its training data $D_a$ for task $T_a$, and access to the pre-trained model $S_{pt}+P_{pt}$ before fine-tuning.
- $\mathcal{K}_{\text{shared}}$: artefacts made visible by the PMM protocol. In particular, the merged shared part, $S_m = \mathcal{M}(S_1, \ldots, S_N)$, is available to participating clients as a white box. Additionally, $C_a$ can query the victim's full model, $S_v+P_v$, as a black box without knowing its model structure and parameters. [3] [4]
- $\mathcal{K}_{\text{aux}}$: optional side information beyond the above. We regard it as the composed clone function:

$$\tilde{f}_v^\star(x) = f(x; S_v, P_v^*), \quad (5)$$

which is structured along three auxiliary axes, determined by (i) white-box access to $S_v$ (e.g., available if $C_a$ is a merging entity, or released via partial open-source); (ii) a structural prior $M_{P_v}$ about $P_v$ where the model parameters are unknown, and (iii) a subset $\hat{D}_v \subset D_v$ of victim data, where $|\hat{D}_v| = p_d \times |D_v|$ and $p_d$ is the proportion of $C_v$'s training data available to $C_a$.

The first two categories, $\mathcal{K}_{\text{self}}$ and $\mathcal{K}_{\text{shared}}$, are *protocol-compliant* and typically available to any honest PMM participant. Our analysis therefore centres on $\mathcal{K}_{\text{aux}}$, systematically varying the availability of $\left(S_v,\ M_{P_v},\ \hat{D}_v\right)$ because these three components directly parameterise the clone $\tilde{f}_v$. For clarity, we encode the presence of these auxiliary axes via indicators:

$$\mathbb{I}_s = \begin{cases} 0 & S_v \text{ is unknown} \\ 1 & S_v \text{ is known} \end{cases},$$

$$\mathbb{I}_p = \begin{cases} 0 & M_{P_v} \text{ is unknown} \\ 1 & M_{P_v} \text{ is known} \end{cases},$$

$$\mathbb{I}_d = \begin{cases} 0 & D_v \text{ is unknown} \\ 1 & D_v \text{ is known} \end{cases}.$$

We identify eight attack scenarios based on the varying levels of $\mathcal{K}_{\text{aux}}$ available to the adversary to examine how different degrees of information exposure affect the feasibility and effectiveness of potential attacks. For simplicity, we

denote the attack scenarios as $\mathcal{AS}[\mathbb{I}_s \cdot \mathbb{I}_p \cdot \mathbb{I}_d]$ for the rest of this paper. For example, $\mathcal{AS}[000]$ means that none of $S_v$, $M_{P_v}$, or $D_v$ is known to the adversary.

### 3.2.2. ADVERSARY'S OBJECTIVE

We define the model accuracy as the performance achieved on task $T_v$ using the validation set.

- **Local accuracy** is the model accuracy of $C_v$'s local pre-merged fine-tuned $S_v + P_v$ (i.e., target model). It serves as the **upper bound** of the model's accuracy as it reflects the performance of the model fine-tuned solely on $T_v$.
- **Merged accuracy** is the model accuracy of the partially-merged model $S_m + P_a$. This accuracy serves as the **lower bound** of the model's accuracy. Note that the merged model is a multi-task model, which is expected to yield lower performance than the fine-tuned single-task models on $T_v$ due to interference between different tasks.
- **Clone accuracy** is the model accuracy of the full clone model $S_v + P_v^*$. It is the **realised accuracy** that directly measures the effectiveness of the model clone attack. The clone model is a single-task model dedicated to $T_v$. Therefore, the clone accuracy is expected to be higher than the merged accuracy.

$C_a$'s objective is to construct a clone private part $P_v^*$ such that the clone accuracy is as high as possible. This would improve the model accuracy that $C_a$ can achieve on $T_v$ using the full clone model $S_v + P_v^*$ or $S_m + P_v^*$, compared to the cases where $C_a$ performs task $T_v$ using $S_v + P_a$ or $S_m + P_a$. [5]

### 3.3. Training the clone model

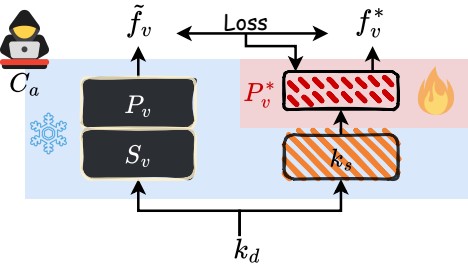

*Figure 5.* Clone model training. $P_v + S_v$ is a black box, $k_s \in \{S_v, S_m\}$ is a frozen white box, and only the cloned private part $P_v^*$ is trained during the attack using $k_d \in \{\hat{D}_v, \hat{D}_a\}$.

We present the overall training procedure of the clone model in Figure 5. The adversary $C_a$ composes a clone model by

---

[3] A practical example of this black-box attack is that the adversary queries a **commercially available model** and uses the responses to reconstruct the proprietary model parameters (Krishna et al., 2020). In this case, after a limited number of queries and a model extraction process, the adversary can maintain their own copy of the model and use it without incurring any further costs to the original model owner.

[4] Note that, different from the conventional black-box attacks, the adversary in the *ModelPirate* attack has additional prior knowledge and can therefore construct a clone model with a better performance depending on the PMM setup, which we will discuss in the following sections of this paper.

[5] Note that the validation set and the training set have no overlapping data samples. The validation set is unknown to $C_a$, and the clone model $P_v^*$ is unknown to $C_v$. Therefore, we measure the clone accuracy to evaluate the attack success rate, which cannot be assessed by either $C_a$ or $C_v$.

*freezing* a shared part $k_s$ and optimising only the private part $P_v^*$ on data $k_d$, thus:

$$\tilde{f}_v^\star(k_d) := f(k_d; k_s, P_v^*),$$
$$k_d \in \{\hat{D}_v, \hat{D}_a\}, \quad k_s \in \{S_v, S_m\}. \tag{6}$$

**For shared part $k_s$ of clone model**. We treat $k_s$ as a *frozen* module, i.e., $\nabla_{\theta^{1:l}} f(k_d; k_s, \cdot) = \mathbf{0}$. The choice of $k_s$ depends on whether the victim's shared part is available ($\mathbb{I}_s \in \{0, 1\}$). Specifically,

- $\mathbb{I}_s = 0$: $C_a$ has no direct knowledge of the victim's shared model part. In this case, the merged shared model $S_m$ is the only model part that embeds task-specific knowledge for $T_v$, and thus we set $k_s = S_m$.
- $\mathbb{I}_s = 1$: under fully distributed merging, $C_a$ can obtain the victim's shared model part $S_v$. Since $S_v$ encodes $T_v$ without task-interference from other clients, it is prioritised over $S_m$, and thus we set $k_s = S_v$.

**For private part $P_v^*$ of clone model**. $P_v^*$ is regarded as the *trainable* module, i.e., $\nabla_{\theta^{1:l}} f(k_d; k_s, P_v^*) \neq \mathbf{0}$. The structure of $P_v^*$ depends on whether the victim's private-structure is known ($\mathbb{I}_p \in \{0, 1\}$):

- $\mathbb{I}_p = 1$: $C_a$ knows $M_{P_v}$ and $P_v$ is architecturally homogeneous with $P_a$ and $P_{pt}$. We initialise $P_v^*$ from the pre-trained parameters $P_{pt}$ to accelerate convergence and better preserve the inductive bias of $P_v$.
- $\mathbb{I}_p = 0$: $C_a$ lacks knowledge of $M_{P_v}$. To mitigate overfitting under limited training data while retaining sufficient model capacity, we adopt a *deep–shallow* design described below.

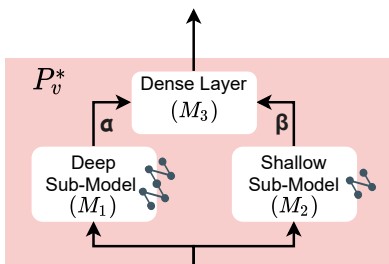

*Figure 6.* Internal structure of $P_v^*$ for $\mathbb{I}_p = 0$. $\alpha$ and $\beta$ are the weight of models $M_1$ and $M_2$, respectively.

For $\mathbb{I}_p = 0$, we assume that $P_v$'s model structure is relatively complex, and a model with a structure similar to $P_v$ can be trained to capture the behaviour of $P_v$. Therefore, we construct a deep sub-model $M_1$ to ensure that the clone model has sufficient complexity to simulate the behaviour of $P_v$. In parallel, a shallow sub-model $M_2$ bypasses $M_1$ to avoid overfitting and enhance generalisation, as some $C_v$'s private training data is unseen by $C_a$.

As shown in Figure 6, the inputs of $P_v^*$ are also the inputs of both $M_1$ and $M_2$, and a dense layer $M_3$ connects the concatenated $M_1$ and $M_2$ outputs to the outputs of $P_v^*$. Depending on $C_a$'s prior knowledge of $C_v$'s model, the deep sub-model $M_1$ can leverage any deep model structure that behaves similarly to $P_v$ (e.g., LSTMs to simulate transformers) with a similar number of layers and neurons. For the worst-case scenario that $C_a$ has zero knowledge about $C_v$'s model structure, $M_1$ should follow a similar structure to $P_a$.

To evaluate the cloned model's task-specific performance, we connect the classification head for task $T_v$ to the output of $P_v^*$. Note that the classification head cannot be merged, and it remains unchanged after model merging. Therefore, the black-box victim model output that the adversary aims to clone is the output of the last layer before the classification head. [6]

**For training data $k_d$ on clone model**. Inputs $k_d$ are fed to both $k_s + P_v^*$ and the black-box of $S_v + P_v$. We denote their outputs as $\tilde{f}_v^*$ and $f_v$, respectively. The choice of $k_d$ is determined by the data-availability indicator $\mathbb{I}_d \in \{0, 1\}$:

$$k_d = \begin{cases} D_a, & \text{if } \mathbb{I}_d = 0 \quad \text{(no victim samples available)}, \\ \hat{D}_v, & \text{if } \mathbb{I}_d = 1 \quad \text{(use victim subset } \hat{D}_v \subset D_v \\ & \qquad\qquad \text{with } |\hat{D}_v| = p_d \times |D_v|). \end{cases} \tag{7}$$

When $\mathbb{I}_d = 1$, $\hat{D}_v$ is prioritised as it is directly aligned with $T_v$.

**Optimisation of $P_v^*$**. The adversary's objective is to align behaviours of $k_s + P_v^*$ and $S_v + P_v$ on the same inputs. We optimise only $P_v^*$, keeping $k_s$ frozen, by minimising a pointwise discrepancy between outputs. We adopt MAE in this paper for its simplicity and effectiveness. [7]

$$\mathcal{L}_{atk} = \frac{\sum_{i \in k_d} \left| \tilde{f}_v^*(i; k_s, P_v^*) - f_v(i, S_v, P_v) \right|}{\left| \tilde{f}_v^*(i; k_s, P_v^*) \right|}. \tag{8}$$

## 4. Experiments

### 4.1. Experimental setup

Unless otherwise specified, we use the default hyperparameters listed in Appendix B for the experiments. For consistency, we consider the layer-wise Task Arithmetic (Ilharco et al., 2023) as the default PMM algorithm in this paper and use the corresponding datasets and models for evaluation. Table 2 in Appendix A shows that when 75% of the layers are merged, the difference between FMM and PMM is less than 10%. Therefore, we set the default proportion of the

---

[6]The classification head for task $T_v$ is used solely to evaluate the task-specific performance of the cloned model. Access to the classification head is not required during clone model training.

[7]Other metrics, such as CE and KL, can also be used.

PMM merged layer to 75%. In addition to the default image classification datasets and vision transformer models used in model merging (Ilharco et al., 2023), we extend our experiments to Natural Language Processing (NLP) tasks using the IMDB (Maas et al., 2011) and QASC (**?**) datasets with the T5 and LLaMA-1B models to show the generalisation of the *ModelPirate* attack beyond the previously considered computer vision models and datasets. The setup for the extended experiments will be detailed in Appendix C.

**Dataset.** Table 3 in Appendix D lists the datasets we consider for model merging. While models fine-tuned on different datasets are used for merging, we will focus on the DTD and EuroSAT datasets as their input features share similar properties (i.e., patterns of different textures and landscapes) while the classification tasks and difficulties differ. We repeat the attack simulations with MNIST and SVHN datasets and present the results in Appendix E. We note that the number of data samples varies across datasets. Therefore, for a fair comparison, we ensure that the number of data samples per class (i.e., see Avg. values in Table 3, Appendix D) is similar across all datasets by randomly selecting a subset of data samples in the "Original" dataset as the "Adjusted" dataset.

**Model.** We consider the Contrastive Language-Image Pretraining (CLIP) (Radford et al., 2021) model with a Vision Transformer (ViT) as the image encoder and a Transformer-based text encoder, following the same model structures in previous model merging literature (Radford et al., 2021; Ilharco et al., 2023), namely ViT-B/32, ViT-B/16, and ViT-L/14. The three pre-trained models are fine-tuned on the datasets listed in Table 3, using the default setups in (Ilharco et al., 2023).

### 4.2. Experimental results

We present the results evaluated on the DTD and EuroSAT datasets. Unless otherwise specified, we denote the clone models as $D_a \rightarrow D_v$, where $D_a$ and $D_v$ are the datasets for fine-tuning the source and target models, respectively. To reduce the impact of outliers while ensuring reproducibility, we repeat the experiments with five different random seeds and present the average values as the results.

#### 4.2.1. OVERALL PERFORMANCE

For benchmark comparison, we consider existing state-of-the-art query-based model clone attacks that match our attack scenarios, namely Knockoff (Orekondy et al., 2019), JBDA (Papernot et al., 2017), and Random (Roberts et al., 2019). The three existing model clone attacks were designed for attack scenarios similar to those described in $\mathcal{AS}$[100], $\mathcal{AS}$[101], and $\mathcal{AS}$[110], respectively. Table 1 lists the accuracies of clone models derived using *ModelPirate* under different attack scenarios $\mathcal{AS}$[XXX] and target model struc-

tures. Note that the default setting requires only about 100 queries to perform attacks.

From the results, we see that in most of the cases where there are at least two of $\mathbb{I}_s$, $\mathbb{I}_p$, or $\mathbb{I}_d$ present, the clone accuracy for *ModelPirate* exceeds or is close to the merged model accuracy. Generally, a simpler target task (i.e., EuroSAT) yields higher clone accuracy, whereas a more complex target task with more classification classes (i.e., DTD) yields lower clone accuracy. From these observations, we conclude that our proposed *ModelPirate* attack substantially outperforms the existing baselines under the same attack scenarios, target model structures, and tasks. Generally, *ModelPirate* attack is **more effective for a less complex target task**.

Interestingly, for ViT-B/32 and ViT-L/14 models, $\mathcal{AS}$[110], where the victim's exact model structure is unknown to the adversary, outperforms $\mathcal{AS}$[111], with the adversary having the same target model structure as the victim and full prior knowledge on the victim's shared model parameters and partial training data. This shows the **advantage of $P_v^*$ we constructed** in Figure 6 compared to the original model structure of the private part when the target model structure is relatively simple. We further explore this observation in the following experiments.

The increase in clone accuracy from $\mathcal{AS}$[X01] to $\mathcal{AS}$[X10] shows that knowing part of the victim's training samples would help the adversary to gain more in its attack performance than knowing the exact model structure of the victim's private part. Therefore, under the default settings, it is **more important for a client to protect its training data than its private model structure**.

The significant increase in clone accuracy from $\mathcal{AS}$[01X] to $\mathcal{AS}$[10X] shows that knowing the victim's shared model part would help the adversary to improve its attack performance more than knowing the victim's private model structure. Therefore, under the default settings, it is **more important for a client to protect its shared model part than its private model structure**. It is suggested that a client should send its shared model part to a trusted merging entity to protect its model privacy.

We also observe that for a more complex model, the clone accuracy increases from $\mathcal{AS}$[1X0] to $\mathcal{AS}$[0X1]. The results show that knowing the victim's shared model part would help the adversary to improve its attack performance more than knowing the victim's training data, because of a large volume of information embedded in the complex model. Therefore, for a more complex model, it is **more important for a client to protect its shared model part than its training data**. For these cases, we suggest that a client should send its shared model part to a trusted merging entity to protect its model privacy. On the other hand, if the victim's model is simpler, an adversary can clone a model with

*Table 1.* Model accuracies of the clone models generated by our proposed *ModelPirate* attacks with the default hyper-parameters, with benchmark comparison with Knockoff, JBDA, and Random model clone techniques. Note that the target model ViT-L/14 is more complex than ViT-B/16, which is more complex than ViT-B/32. Bolded numbers indicate successful model clone attacks in which the clone accuracy exceeds the merged accuracy.

| Attack Method | EuroSAT→DTD | | | DTD→EuroSAT | | |
|---|---|---|---|---|---|---|
| | ViT-B/32 | ViT-B/16 | ViT-L/14 | ViT-B/32 | ViT-B/16 | ViT-L/14 |
| **Merged Acc.** | 52.77% | 54.84% | 72.50% | 67.67% | 79.11% | 95.04% |
| Knockoff | 7.18% | 7.18% | 2.13% | 52.22% | 57.22% | 16.70% |
| JBDA | 3.88% | 3.56% | 3.19% | 23.48% | 29.33% | 18.63% |
| Random | 2.93% | 2.13% | 35.37% | 10.44% | 12.67% | 38.70% |
| $\mathcal{AS}$[000] | 2.39% | 2.45% | 1.91% | 15.74% | 17.89% | 14.74% |
| $\mathcal{AS}$[100] | 37.27% | 8.54% | 2.44% | 53.74% | 31.07% | 23.61% |
| $\mathcal{AS}$[010] | 2.66% | 2.45% | 65.37% | 48.52% | 54.70% | **93.81%** |
| $\mathcal{AS}$[001] | 18.88% | 26.22% | 20.53% | 60.07% | 60.04% | 59.67% |
| $\mathcal{AS}$[101] | **68.35%** | **60.79%** | 31.23% | **96.83%** | **95.81%** | 79.37% |
| $\mathcal{AS}$[011] | 49.89% | **56.81%** | **82.82%** | 65.52% | 68.22% | 73.37% |
| $\mathcal{AS}$[110] | **85.15%** | **78.40%** | **97.87%** | **98.38%** | **98.93%** | 52.54% |
| $\mathcal{AS}$[111] | **62.89%** | **65.42%** | **98.19%** | **96.34%** | **96.52%** | 63.22% |

better performance using a subset of the victim's dataset, even without any knowledge of the victim's public model. Therefore, **protecting the training data is more important for a client to reduce model privacy leakage**.

#### 4.2.2. PRIVACY-UTILITY TRADEOFF: IMPACT OF THE NUMBER OF MERGED LAYERS

Figure 2 in Section 1 shows that in a partially-merged setup, the merged model accuracy for each individual task increases as the number of layers merged increases (i.e., a larger $l$). However, this comes at the cost of reduced privacy, as the difficulty of reconstructing the behaviour of the model part intended to remain private increases, because the adversary can obtain a larger volume of information. In this experiment, we vary the separation layer $l$ and set the remaining parameters to their default values to demonstrate this hypothesis qualitatively and quantitatively. We repeat the experiment for $\mathcal{AS}$[101] and $\mathcal{AS}$[111] and present the results as "Known Model" and "Unknown Model" correspondingly.

The experimental results in Figure 7 show that the general trends of clone model accuracies under the assumptions of known or unknown target model structure increase as the number of merged layers (i.e., a larger $l$) increases. We also see that the increase in the clone model accuracy is more significant in the EuroSAT→DTD scenario, where the target model performs a more difficult task than the source model.

As shown in Figure 7 (top), the clone accuracy is less than the merged model accuracy when $l < 7$. The target model's clone accuracy surpasses the merged model accuracy at $l = 7$. Similarly, Figure 7 (bottom) shows the DTD→EuroSAT scenario where the source model performs a more difficult

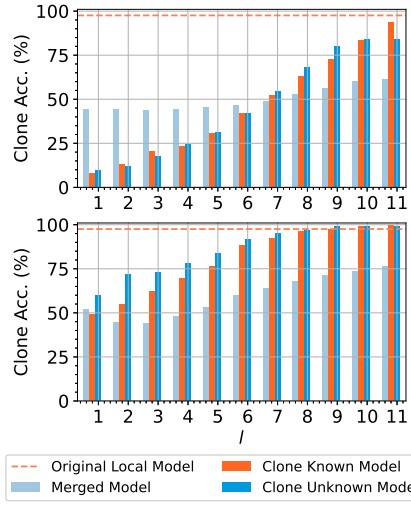

*Figure 7.* Clone accuracy for different numbers of merged layers $l$ in EuroSAT→DTD (top) and DTD→EuroSAT (bottom) scenarios.

task than the target model. The clone accuracy under this scenario is always higher than the merged model accuracy at the same separation layer, except at layer one, where the known model clone accuracy is slightly lower than the merged model accuracy. Interestingly, we see that the clone accuracies for the last few layers (i.e., $l > 7$) are close to or even greater than the original local model's accuracy. This shows that merging more than seven layers would create a significant privacy vulnerability in the private part's model behaviour. We also conclude that the model clone attack is more successful when it clones a target model for a less difficult task.

### 4.2.3. IMPACT OF THE PROPORTIONS OF KNOWN DATA SAMPLES

Next, we focus on how the proportion of data samples the adversary can obtain from the victim affects the clone accuracy. We use subsets of the data samples from the "Adjusted" dataset (see Table 3) to ensure that the total number of data samples from each class is similar.

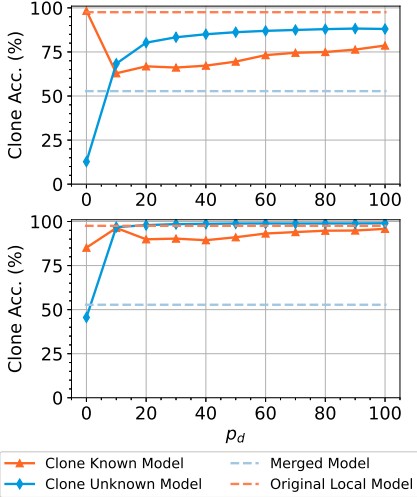

*Figure 8.* Clone accuracy for different proportions of data samples ($p_d$) in EuroSAT→DTD (top) and DTD→EuroSAT (bottom) scenarios.

Figure 8 shows that from when $p_d$ is between 20% and 100%, the increase in $p_d$ results in an increase in the clone model accuracy for both known and unknown models. The clone model accuracy surpasses the merged model accuracy for all cases at $p_d = 10\%$. Note that in this experiment, the cases with $p_d = 0\%$ belong to $\mathcal{AS}[100]$ and $\mathcal{AS}[110]$ for unknown and known model scenarios, respectively. For those cases, the adversary trains the clone model on its own dataset. Therefore, we observe that the clone model accuracy under the known model scenario (i.e., $\mathcal{AS}[110]$) is higher than that under the unknown model scenario (i.e., $\mathcal{AS}[100]$). This is because the model is randomly initialised in $\mathcal{AS}[100]$, whereas the initial model in $\mathcal{AS}[110]$ is the pre-trained model with higher accuracy on the target task. The merged model accuracy remains consistent for all $p_d$, as it only depends on the number of layers merged, given the same fine-tuned models.

## 5. Related work

Model merging (aka model fusion) is a technique that combines the parameters of models with different capabilities to build a single multi-task model (Yang et al., 2024a). A typical approach to construct a multi-task model using the model merging technique is to use a common pre-trained model as a backbone and merge the fine-tuned models for different downstream tasks (Matena & Raffel, 2022; Ilharco et al., 2023; Yadav et al., 2023). To reduce resource consumption in conventional FMM, PMM (Stoica et al., 2024; Wang et al., 2025) was proposed to merge only a subset of the parameters or layers in a model. However, previous empirical studies on PMM mainly focused on resource reduction and performance optimisation, and the privacy protection perspective of PMM remains unexplored.

Model clone is a group of privacy attacks targeting NN models in which the adversary aims to construct an alternative NN model that behaves similarly to the victim model. Papernot et al. (2017) proposed a model clone attack based on the assumption that the adversary can access a subset of training data, but the model structure is unknown to the adversary. The adversary trains an alternative model with decision boundaries similar to those of the victim's model using a synthetic dataset (Papernot et al., 2017). The dataset is generated based on the accessible subset of the data using a technique named Jacobian-based Dataset Augmentation (JBDA) (Papernot et al., 2017). Alternatively, Orekondy et al. (2019) proposed a model-clone attack model where the adversary aims to steal the functionalities of the victim model. Their technique, named "Knockoff" (Orekondy et al., 2019), is based on a black-box assumption similar to (Papernot et al., 2017). However, the adversary in (Orekondy et al., 2019) cannot access any of the training data samples, and an alternative set of data is used to train the "Knockoff" model. A similar data-free model extraction attack is presented by Biton Dor & Mirsky (2024). Roberts et al. (2019) showed that it is also possible to perform model clone attacks using only randomly generated data samples, given that the adversary has knowledge of the victim model's structure. While some early works identified privacy risks in model merging (Lu et al., 2025) or partially black-box API queries (Carlini et al., 2024), none of the existing model clone attacks is targeted at cloning partially-shared models' behaviour during model merging.

## 6. Conclusion

In this paper, we proposed and analysed a model clone attack in PMM. The adversary can perform the attack with different prior knowledge, including the victim's shared model parameters, private model structure, and training data samples. We performed attack simulations to compare our proposed attacks with existing model clone attacks, with the same assumption about the adversary's prior knowledge. We showed that our attack is more successful than the baseline attacks in most of the scenarios we considered. We also explored our proposed attack with various numbers of private layers and data leakage and formalised a layer selection process in Appendix H. Results show that keeping fewer

layers private can improve the merged model's performance at the cost of a higher attack success rate. Only a small fraction of data leakage can help the adversary achieve a better attack performance. Furthermore, we showed that the adversary can leverage a deep-shallow model structure to simulate the behaviour of an unknown model with similar or higher performance compared to the original model.

## Acknowledgements

This research is supported by the RIE2025 Industry Alignment Fund–Industry Collaboration Projects (IAF-ICP) (Award I2301E0026), administered by A*STAR, as well as supported by Alibaba Group and NTU Singapore through Alibaba-NTU Global e-Sustainability CorpLab (ANGEL). This research is also supported by the Ministry of Education, Singapore, under its Academic Research Fund Tier 2 (Award MOE-T2EP20125-0005).

## Impact Statement

This paper presents work whose goal is to advance the field of Machine Learning. There are many potential societal consequences of our work, none which we feel must be specifically highlighted here.

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

## A. Model accuracies for ViT-B/32, ViT-B/16 and ViT-L/14 models

Table 2 shows that the three models fine-tuned on the DTD and EuroSAT datasets achieve more than 97.5% model accuracy on their respective datasets. However, the model accuracy degrades sharply to 56.3% on the alternative dataset (i.e., DTD→EuroSAT or EuroSAT→DTD). This indicates that the cross-task model performance is poor for the pre-merged fine-tuned model optimised for a specific task. The lack of generalisation performance motivates the need for model merging, which combines the capabilities of multiple single-task models into a single unified multi-task model without additional fine-tuning.

After merging all single-task models, the resulting multi-task model achieves up to 95.89% (with FMM) or 95.04% (with PMM), substantially outperforming the individual single-task models. Notably, we observe that when 75% of the model parameters are merged, PMM still yields at least 85.63% model accuracy related to FMM. The results demonstrate robust performance retention under partial merging.

*Table 2.* Model accuracies of the fine-tuned models, fully-merged model (FMM), and partially-merged model with 75% merged layers (PMM).

|                       | DTD      |          |          | EuroSAT  |          |          |
| --------------------- | -------- | -------- | -------- | -------- | -------- | -------- |
| Model                 | ViT-B/32 | ViT-B/16 | ViT-L/14 | ViT-B/32 | ViT-B/16 | ViT-L/14 |
| Parameters per Layer  | 7087872  | 7087872  | 12596224 | 7087872  | 7087872  | 12596224 |
| Fine-tuned (DTD)      | **97.55%** | **98.14%** | **98.24%** | 35.00% | 34.07% | 56.30% |
| Fine-tuned (EuroSAT)  | 34.52%   | 35.53%   | 47.61%   | **99.85%** | **99.89%** | **99.93%** |
| FMM                   | 61.44%   | 64.04%   | 77.87%   | 76.41%   | 79.67%   | 95.89%   |
| PMM                   | 52.77%   | 54.84%   | 72.50%   | 67.67%   | 79.11%   | 95.04%   |
| PMM/FMM               | 85.89%   | 85.63%   | 93.10%   | 88.56%   | 99.30%   | 99.11%   |

Similar to Figure 2, Figures 9a and 9b show that the accuracy of the merged model generally increases as more layers are merged across all five datasets for ViT-B/16 and ViT-L/14 models.

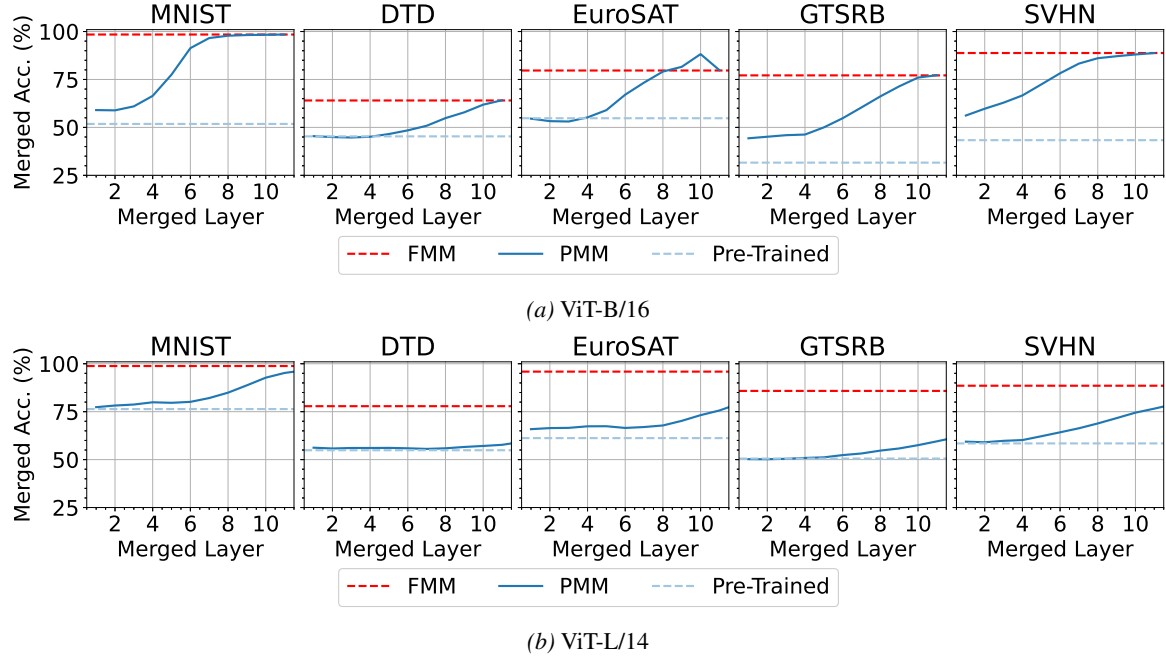

*Figure 9.* Merged model accuracy for the partially merged model with different numbers of merged layers. The merging is performed between models fine-tuned on five different downstream tasks (i.e., MNIST, DTD, EuroSAT, GTSRB, and SVHN).

## B. Default experimental settings

We use a workstation equipped with an Intel Xeon Gold 6248R CPU and two NVIDIA RTX A5000 GPUs. The memory size is 128 GB. The NN model training and merging are based on the PyTorch Python library. We set the number of training rounds to be 1500 after performing some trial runs to ensure that the model has converged by round 1500.

Unless otherwise stated in the experimental results, we use the following default hyperparameters:

- Target model structure: ViT-B/32
- The victim's shared model part is known: $\mathbb{I}_s = 1$
- Known data (i.e., $\mathbb{I}_d = 1$): 10% of the data samples in the adjusted dataset (i.e., $p_d = 10\%$);
- Sub-model weights for $\mathbb{I}_p = 0$: $\alpha = \beta = 0.5$;
- Deep sub-model (i.e., $M_1$) for $\mathbb{I}_p = 0$: multilayer LSTM;
- Learning rate for the clone model: 1e-5 for $\mathbb{I}_p = 0$ (deep-shallow model), 0.001 for $\mathbb{I}_p = 1$ (original model structure);

## C. Results on the T5 and LLaMA-1B model

To assess the generalisation, we extend our evaluation for the *ModelPirate* attack beyond vision transformers and classification tasks using the T5-based encoder–decoder large language model (LLM) architecture (Raffel et al., 2020), which differs substantially in structure and complexity from the previously evaluated encoder-only vision transformers for image classification tasks. Specifically, we conducted experiments on two NLP tasks – sentiment analysis on the IMDB dataset (Maas et al., 2011) and question answering on the QASC dataset (Khot et al., 2020). The fine-tuned models are publicly available at Hugging Face [8] [9].

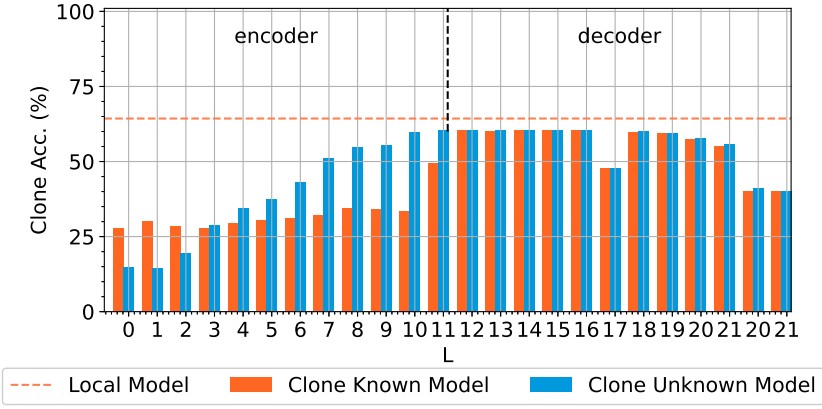

*Figure 10.* Clone accuracy for different layers in T5 model.

In Figure 10, we present the results using IMDB sentiment analysis as the source task and QASC question answering as the target task. We see that the trend in the encoder's clone accuracy is similar to that observed for the ViT models. Compared with the encoder, the decoder module yields consistently higher clone accuracy. The reason is that the decoder expands and transforms the information already extracted by the encoder. This makes the information bottleneck lie between the encoder and the decoder, such that no additional information can be added to the encoder's outputs.

The ViT model used in the main experiments is an encoder-only model with the classification head directly attached to the encoder output. The behaviour of the ViT model is therefore expected to be similar to the encoder part of the T5 model, where layers closer to the outputs extract more task-specific features. The results confirm this expectation and demonstrate that *ModelPirate* can effectively replicate the target model's behaviour and construct a clone model with similar model accuracy to the target model, especially for $l$ between 12 and 16 within the decoder module of the T5 model.

In addition, we have performed experiments with the LLaMA-1B decoder-only model using the IMDB dataset. Results show that the clone model accuracy achieves 94.90% in the unknown dataset scenario and 95.58% in the partially known

---

[8]https://huggingface.co/mrm8488/t5-base-finetuned-imdb-sentiment
[9]https://huggingface.co/mrm8488/t5-base-finetuned-qasc

dataset setting. The results are close to the official reported accuracy of the LLaMA-1B model fine-tuned on the IMDB dataset [10], which is 96.04%.

## D. Datasets used for the experiments

Five datasets are selected to cover a broad range of image classification tasks. The tasks chosen include: handwritten digit classification using the MNIST dataset (Lecun et al., 1998), textural classification using the DTD dataset (Cimpoi et al., 2014), land use and land cover classification using satellite remote-sensing images with the EuroSAT dataset (Helber et al., 2019), traffic sign classification using the GTSRB dataset (Stallkamp et al., 2012), and house number classification from street photos with the SVHN dataset (Netzer et al., 2011). The datasets used for the main experiments are listed in Table 3.

*Table 3.* Datasets used for the main experiments.

| Dataset | Classification Task | Classes | Original | | Adjusted | |
| --- | --- | --- | --- | --- | --- | --- |
| | | | Samples | Avg. | Samples | Avg. |
| MNIST (Lecun et al., 1998) | Handwritten digits | 10 | 60000 | 6000 | 2000 | 200 |
| DTD (Cimpoi et al., 2014) | Textural image | 47 | 5640 | 120 | 5640 | 120 |
| EuroSAT (Helber et al., 2019) | Land use and cover | 10 | 27000 | 2700 | 2700 | 270 |
| GTSRB (Stallkamp et al., 2012) | Traffic sign | 43 | 51840 | 1206 | 5184 | 121 |
| SVHN (Netzer et al., 2011) | House number digits | 10 | 99289 | 9929 | 3310 | 331 |

## E. Experimental results for MNIST and SVHN datasets

In previous experiments, we only considered a pair of source and target tasks (i.e., DTD and EuroSAT). We repeat these experiments for an alternative pair of source and target tasks (i.e., MNIST and SVHN). Specifically, Figure 11 compares model clone attack effectiveness at different layers, and Figure 12 compares attacks when varying the proportions of known data samples. These results show a similar trend to the results in the previous sections, highlighting the robustness of *ModelPirate* across diverse attack setups with various dataset pairs.

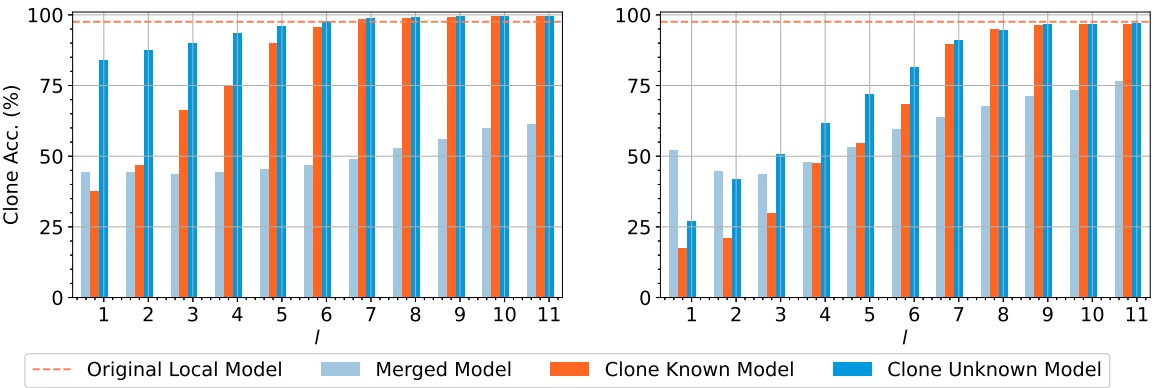

*Figure 11.* Clone accuracy for different numbers of merged layers $l$ in MNIST→SVHN (left) and SVHN→MNIST (right) scenarios.

## F. Experimental results for $\mathbb{I}_p = 0$ with different $P_v^*$ sub-model structures

In previous experiments, we only considered a pair of source and target tasks (i.e., DTD and EuroSAT). Then, we further expand the experiments for $\mathbb{I}_p = 0$ for all source and target tasks with the rest of the parameters set as defaults. To analyse the impact on the clone model accuracy by the deep and shallow sub-models (i.e., $M_1$ and $M_2$ in Figure 6), we remove $M_1$ or $M_2$ and repeat the simulation subsequently. From Table 4, we see that the clone model $P_v^*$ with all sub-models $M_1 + M_2 + M_3$ yields higher clone accuracy than $M_1 + M_3$, and similar clone accuracy as $M_2 + M_3$. Based on this observation, we conclude that the shallow model $M_2$ contributes more to the clone model's accuracy than the deep model

---

[10]https://huggingface.co/yash3056/Llama-3.2-1B-imdb

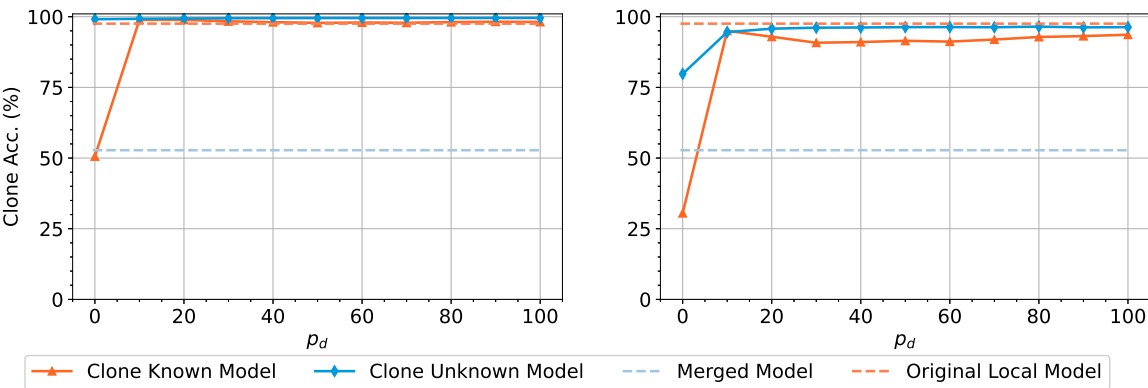

*Figure 12.* Clone accuracy for different proportions of data samples ($p_d$) in MNIST→SVHN (left) and SVHN→MNIST (right) scenarios.

$M_1$.

*Table 4.* Clone Accuracy for different $P_v^*$ model structures and target tasks.

*(a) $\mathcal{AS}$[100]*

|  | MNIST | DTD | EuroSAT | GTSRB | SVHN |
|---|---|---|---|---|---|
| $M_1 + M_2 + M_3$ | 99.20% | 67.93% | 97.37% | 90.87% | 94.66% |
| $M_2 + M_3$ | 99.21% | 67.18% | 97.41% | 91.09% | 94.85% |
| $M_1 + M_3$ | 97.99% | 55.32% | 87.67% | 76.29% | 85.91% |

*(b) $\mathcal{AS}$[101]*

|  | MNIST | DTD | EuroSAT | GTSRB | SVHN |
|---|---|---|---|---|---|
| $M_1 + M_2 + M_3$ | 42.57% | 5.11% | 33.31% | 11.67% | 49.25% |
| $M_2 + M_3$ | 42.19% | 5.32% | 34.14% | 13.40% | 49.78% |
| $M_1 + M_3$ | 30.36% | 3.66% | 24.09% | 5.15% | 29.94% |

To further evaluate the benefit of the deep-shallow design, we replace the deep-shallow module with a single fully-connected layer while keeping all other settings unchanged.

*Table 5.* Clone Accuracy for the deep-shallow and fully-connected model structures and target tasks.

|  | MNIST | DTD | EuroSAT | GTSRB | SVHN |
|---|---|---|---|---|---|
| Deep-shallow | 97.5% | 54.2% | 93.6% | 46.1% | 60.3% |
| Fully-connected | 99.1% | 14.2% | 80.1% | 15.5% | 40.6% |

Results in Table 5 show that replacing the deep-shallow module with a simple fully-connected layer substantially reduces attack efficacy (up to 40%) on most datasets considered. This provides additional evidence that the DS module plays an important role in the effectiveness of the attack.

## G. Fine-grained layers in a ViT residual block

In previous experiments, we only considered separating the model after a residual block in the ViT model. We conduct experiments to investigate how merging different components within a ViT residual block affects the effectiveness of *ModelPirate*. We repeat the experiments for separating after the attention block, feed-forward block, and the entire residual block at $l = 7$. As shown in Figure 13, the attack remains effective across all configurations for two benchmark tasks, with nuanced variations in clone accuracy depending on which sub-component is shared.

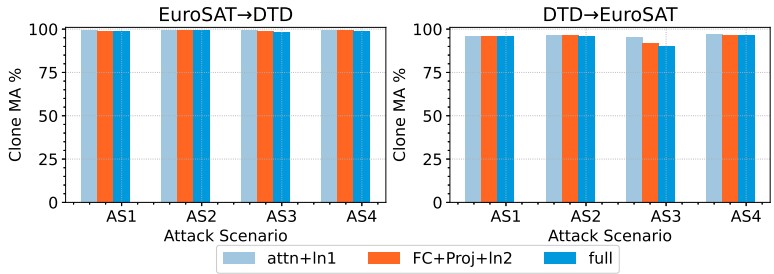

*Figure 13.* Clone accuracy for different fine-grained layers in ViT residual block eight.

## H. Layer selection guideline

We formalise the layer selection process as an optimisation problem that balances privacy leakage and performance gain. Specifically, a client can determine the optimal number of private layers by minimising a composite objective that incorporates (i) layer-wise information exposure, measured via auxiliary loss on neuron activations, and (ii) performance improvement, quantified by incremental gains across layers. The trade-off is controlled by a user-defined scaling factor. Let the optimal number of private layers be $l^*$. Then, we have

$$\underset{L^*}{arg min} \frac{1}{L} \sum_{l=1}^{L} \left( (1 + l * \epsilon) \times \phi_l - \lambda \Delta p_l \right) \qquad (9)$$

Where:

- $\epsilon$ represents the incremental privacy leakage per layer. Theoretically, the cumulative privacy exposure increases non-linearly as the increase in the number of layers shared due to the additional information embedded in the combined layers compared to individual layers;
- $\phi_l$ is the information carried by all neurons in layer $l$, estimated via an auxiliary loss on the neuron activations. A higher auxiliary loss implies that the activations contain more informative (and potentially sensitive) content;
- $\Delta p_l$ is the performance gain by the layer. It can be computed as the difference in the model performance by re-training a partial model, up to layers $l$ and $l - 1$; and
- $\lambda$ is the scaling factor determined by the clients to balance the privacy loss and performance loss measurements.

## I. Computational and communication overheads

We measure the change in computational and communication overheads with different numbers of layers merged. Let the time consumption for merging $l$ out of $L$ layers be $t_l$. The average increase in computational and communication costs when an additional layer is merged is calculated as:

$$100\% \times \frac{\sum_{l=2}^{L} \frac{t_l - t_{l-1}}{t_{l-1}}}{L - 1}. \qquad (10)$$

We summarise the results in Table 6. The results indicate that each additional layer shared increases the computational and communication costs by approximately 3.83% to 7.23%, depending on the model structure. The results in Table 6, together with Figure 2, demonstrate a trade-off between privacy preservation and model utility.

*Table 6.* Computational and communication costs when an additional layer is merged.

| Model | ViT-L/14 | ViT-B/16 | ViT-B/32 |
|---|---|---|---|
| Overhead | 3.83% | 7.23% | 6.41% |

