# OpenReview forum: "An Empirical Study on the Resilience of Partial Merging to Model Clone Attacks"
_ICML.cc/2026/Conference — ICML 2026 regular_

### Official Review · Reviewer_mEXP · 2026-02-17

**Soundness:** 1
**Presentation:** 2
**Significance:** 1
**Originality:** 2
**Overall Recommendation:** 2
**Confidence:** 4

**Summary:**

This paper investigates the privacy vulnerabilities of Partial Model Merging (PMM), where models are split into shared (merged) and private (local) parts. The authors propose a model clone attack framework to reconstruct the unshared private portions of partially merged models and evaluate it under eight attack scenarios with varying adversarial knowledge (partial training data, model parameters, and/or model structure). Key findings include: (1) PMM remains highly vulnerable to model stealing even with limited adversarial knowledge, (2) a "deep-shallow" clone model architecture (deeper private part, shallower shared part) improves attack success, and (3) clone model accuracy can exceed the original merged model accuracy in some cases. Experiments on MNIST, DTD, EuroSAT, GTSRB, and SVHN with ViT-B/32 demonstrate that adversaries can recover significant portions of private model performance even when only a small fraction of training data or parameters is exposed.

**Compliance With Llm Reviewing Policy:**

Affirmed.

**Key Questions For Authors:**

1. Can you provide citations supporting the claim that PMM emerged in response to model stealing concerns? The cited PMM papers (ZipIt!, etc.) appear to motivate PMM primarily for efficiency rather than privacy. If PMM was not designed with privacy in mind, should the framing be revised?

2. Can you clarify the practical use case for PMM when task heads are kept private? PMM methods like ZipIt! are designed to create a unified multi-task model that can classify inputs across the superset of all merged classes, which requires all task heads to be publicly accessible. If heads remain private with individual clients, what is the advantage of merging shared layers over simply using separate models? How does your threat model align with realistic PMM deployments?

3. How does the efficacy of the attack change if you don't adopt a deep-shallow design?

**Limitations:**

yes

**Strengths And Weaknesses:**

### Strengths

1. The paper is easy to understand.

2. The insight that clone models should mirror the PMM structure to maximize attack success is intuitive.

3. The ablation across different numbers of merged layers, datasets, and knowledge assumptions is well-structured and makes it easy to understand which factors drive attack effectiveness.

### Weaknesses

1. The introduction claims that PMM emerged "in response" to model stealing risks in FMM. However, this is not supported by the PMM literature. The cited ZipIt! paper (Stoica et al., 2024) and other PMM work primarily motivate the approach for computational efficiency and parameter reduction, not privacy. This mischaracterization weakens the paper's framing.

2. Partial model merging seems to be used in settings where you want to combine two classification models so that you can perform classification over the superset of the classes of the two models. I don't understand why partial model merging is useful if you keep the head (last few layers) of one of the models private. The threat model that this paper considers does not make sense in the context of partial model merging.

3. Beyond the deep-shallow clone model architecture, the attack framework primarily adapts standard model stealing techniques (knowledge distillation, data augmentation) to the PMM setting. The contribution is largely empirical—demonstrating that existing attacks work on PMM—rather than introducing novel attack mechanisms. While such empirical studies have value, the methodological contribution is incremental.

---

> ### Author Rebuttal · Authors · 2026-03-31
>
> **W1/Q1: PMM Model Stealing Attack Literatures**
>
> We thank the reviewer for the insightful comment. We agree that prior PMM methods, such as ZipIt!, were not originally proposed as defences against model stealing. However, model privacy is an increasingly important consideration, particularly as proprietary models become more widely commercialised.
>
> Prior work on personalised federated learning (PFE) [2] and split learning (SL) [3,4] suggests that different layers encode different levels of information. Typically, early layers tend to carry more general information, whereas later layers become increasingly task-specific. This suggests that sharing different portions of a model may expose different levels of privacy leakage. Results in [3] show that the difficulty of model stealing attacks increases with the increase in the number of shared model portions.
>
> Although both PFE/SL and PMM involve **partial model sharing**, PMM is distinct because clients share their fine-tuned, task-specific models through **one-shot** model aggregation. As a result, the adversary in PMM observes less information than in iterative collaborative training settings, making model stealing in PMM more challenging and distinct. We therefore see that PMM is a valuable setting in which to study model stealing risks from partial model sharing.
>
> From a defense perspective, the model owner can encrypt a selected subset of layers, balancing computational cost and defense effectiveness. Alternatively, in the PMM, the model owner can keep a selected subset of layers **private instead of encrypting** them. From the adversary’s perspective, the amount of information exposed under partial encryption is expected to be similar to that under PMM, given the same protected layers.
>
> Regarding the amount of model privacy leakage of PMM versus FMM, we provide a simple information-theoretic proof showing that **full-parameter sharing cannot be safer than partial-parameter sharing** with respect to model IP leakage:
>
> - Let the full trained model parameters be $θ=(θ^1,\dots,θ^L)$, shared model parameters be $S(θ)=(θ^1,\dots,θ^l)$. where $\{1,\dots,l\} \subset \{1,\dots,L\}$.
> - $S(θ)$ is obtained by selecting a subset of $θ$ at layer $l$. Therefore, $S(θ)$ is a deterministic function of $θ$. For any model extraction target $T=f(θ)$, this induces the Markov chain: $T\to\theta\to S$.
> - Then, let the mutual information between $T$ and $S(θ)$ be $I(T;S)$ and between $T$ and $\theta$ be $I(T;θ)$. Since $S$ is a post‑processing of $θ$, $S$ should contain less information than $\theta$, according to the data processing inequality.
> - We therefore have $I(T;S)\leq I(T;θ)$, which means that sharing a subset of model parameters cannot increase the mutual information compared to sharing the full model.
>
> According to the above proof, sharing the full model results in more model IP leakage than sharing only a subset of the parameters, thereby making it more vulnerable to model stealing attacks.
>
> [2].Chen et al. (IJCAI 2025). Optimizing personalized federated learning through adaptive layer-wise learning
>
> [3].Erdoğan et al. (2022). Unsplit: Data-oblivious model inversion, model stealing, and label inference attacks against split learning
>
> [4].Li et al. (2023). Model extraction attacks on split federated learning
>
> [5].Jiang et al. (2026). Intellectual property protection for deep learning model and dataset intelligence
>
> **W2/Q2: Practical Use of PMM**
>
> We thank the reviewer for highlighting the need to clarify the practical use case for PMM. We follow the prior work on task vector and ZipIt!, where the classification head is not considered in merging, and it is not explicitly public or private. Our setup is intended to remain consistent with these prior FMM/PMM settings.
>
> In practice, a user can attach or request a classification head for the inference task it intends to perform. Therefore, our analysis does not explicitly assume a specific visibility status for the classification head, in line with previous FMM/PMM works cited.
>
> **W3/Q3: Deep-Shallow Design**
>
> We thank the reviewer for raising this important point. We provide an ablation study in Appendix F that uses only the deep ($M_1$) or the shallow ($M_2$) part of the model. We find that the shallow component contributes more to clone model performance. To further evaluate the benefit of the deep-shallow (DS) design, we replace the DS module with a single fully-connected (FC) layer while keeping all other settings unchanged. The supplementary results on five datasets are reported below:
>
> ||MNIST|DTD|EuroSAT|GTSRB|SVHN|
> |-|-|-|-|-|-|
> |DS|97.5\%|54.2\%|93.6\%|46.1\%|60.3\%|
> |FC|99.1\%|14.2\%|80.1\%|15.5\%|40.6\%|
>
> These results show that replacing the DS module with a simple FC layer substantially reduces attack efficacy (up to ~40\%) on most datasets considered. This provides additional evidence that the DS module plays an important role in the attack effectiveness. We will clarify this point in the revised paper.

---

> > ### Author Rebuttal · Reviewer_mEXP · 2026-04-05
> >
> > I thank the authors for their rebuttal. Unfortunately my main concerns with this paper remain unresolved.
> >
> > W1. My core criticism was about the paper’s framing of prior PMM work. The introduction presents PMM as emerging “in response” to model privacy/model-stealing risks in FMM, but neither the cited PMM literature nor the paper’s own discussion supports that historical framing. The rebuttal does not address this point directly. Instead, the rebuttal presents generic arguments that privacy is an important consideration without showing that there are concrete proposals for using PMM as a privacy preserving mechanism. The comparison with federated and split learning  is tenuous. I’m unsure why the authors present arguments to show that “full parameter sharing cannot be safer than partial parameter sharing” as this was never contested/mentioned in my review.
> >
> > W2. The rebuttal also does not provide a convincing practical motivation for the specific PMM setup studied here as a privacy mechanism. In particular, the paper keeps task-specific heads/private parts separate while otherwise following the ZipIt!-style setup, but does not give concrete deployment examples or prior proposals where this separation is adopted for privacy reasons rather than for efficiency or compatibility reasons. As written, the paper therefore reads more like a security analysis of an efficiency-motivated paradigm than an evaluation of a privacy-motivated defense.

---

> > > ### Author Response · Authors · 2026-04-07
> > >
> > > Thank you for the follow-up. We are glad that Weakness 3 has been resolved, and we appreciate the opportunity to further clarify Weaknesses 1 and 2. In our view, these remaining concerns are primarily matters of framing and exposition rather than technical validity, where **no substantial changes are required**.
> > >
> > > ---
> > >
> > > # Weakness 1
> > >
> > > **Weakness 1** concerns the paper’s wording and historical framing. We agree with the reviewer that the primary motivation in the cited PMM literature (e.g., *ZipIt!*) is efficiency and parameter reduction. To address this concern, **we will remove the claim** in the introduction that PMM is proposed "in response" to model privacy or model-stealing risks in FMM.
> > >
> > > Our intended claim is narrower: **even when PMM is adopted for efficiency and parameter reduction rather than privacy, it induces a distinct partial-disclosure setting that creates a meaningful model-stealing attack surface**. This aligns with the setting studied in our paper. In other words, we do not claim that PMM was previously established as a privacy-preserving mechanism. Instead, we analyze the security consequences of partial parameter exposure in PMM-style deployments and argue that this practical deployment introduces a nontrivial model extraction surface that merits dedicated analysis.
> > >
> > > We believe our work is still grounded in a well-motivated security question. Many systems are introduced for utility and efficiency reasons, and are later analyzed through the lens of privacy or security. Our paper fits that pattern: **PMM changes the volume of model information exposed to an adversary** relative to FMM, so the relevant research question is whether existing model-stealing methods remain effective, and how the attack efficacy varies with the amount of exposed information, as well as with architectural and knowledge assumptions. That is exactly what our study addresses.
> > >
> > > ---
> > >
> > > # Weakness 2
> > >
> > > **Weakness 2** requires clearer practical examples and stronger motivation for deployment. We agree with the reviewer that the current manuscript did not make the practical setup sufficiently concrete. In the revision, we will clarify that our threat model should be understood as a **security analysis of PMM under restricted access to task-specific components**, rather than as an evaluation of a previously established **privacy-motivated PMM defense**.
> > >
> > > In our current manuscript, we decompose the threat model along **three dimensions** of prior knowledge -- shared model, structural prior, and victim data. We conduct model-stealing security analyses across the resulting eight settings. These settings collectively cover a wide range of real-world scenarios with varying degrees of information disclosure.
> > >
> > > Concretely, our intended deployment picture reflects real-world scenarios where a cloud server or merging entity releases a merged backbone feature extractor to reduce communication and computation costs for participants. This mirrors industry practices in Model-as-a-Service (MaaS), where providers allow users to query high-performance models as a "black box" while keeping the proprietary structure and "last-mile" parameters hidden. For example, a consortium might share a high-capacity vision backbone (e.g., ViT-L/14) to benefit all hospitals, while each hospital keeps its local data and final diagnostic layers (private part $P_n$) local to protect proprietary insights. This corresponds to *AS110* in our study. In such settings, task-specific heads are subject to access control, API restrictions, or client-local deployment. The shared merged portion is therefore exactly the artifact exposed to an attacker, and we evaluate how much of the **hidden task-specific functionality** can still be cloned from that partial exposure.
> > >
> > > ---
> > >
> > > # Summary
> > >
> > > In summary, we will revise the paper to make this scope explicit in three ways:
> > >
> > > 1. We will carefully revise our original framing of the paper in the final version and **remove the claim** that PMM historically emerged as a response to model-stealing risk;
> > > 2. We will state clearly that the paper studies **model extraction risk in PMM-style partial-exposure settings**. More specifically, we will clarify that our contribution is framed as a **security analysis of an efficiency-motivated paradigm**, regardless of the original motivation for PMM;
> > > 3. We will include clearer practical examples and concrete real-world case studies to explicitly illustrate the deployment setting captured by our attack scenario.
> > >
> > > Our central contribution of the paper is **PMM-style partial parameter disclosure changes the attacker’s observability, but does not eliminate model-extraction risk; this attack surface therefore deserves explicit analysis**. Importantly, addressing these concerns **does not** require any change to the paper's technical substance, the threat model, or the experimental results.
> > >
> > > We hope that the above clarifications adequately address the main concerns and support a more favorable consideration.

---

### Official Review · Reviewer_dk1o · 2026-03-04

**Soundness:** 3
**Presentation:** 3
**Significance:** 2
**Originality:** 1
**Overall Recommendation:** 4
**Confidence:** 2

**Summary:**

As the title suggests, this paper is an empirical study on the resilience of partial model merging to model clone attacks. It considers eight scenarios and turns an intuitive privacy hypothesis into a testable investigation.

**Compliance With Llm Reviewing Policy:**

Affirmed.

**Final Justification:**

My assessment of the strengths and weaknesses remains the same as in my original review.

The authors’ rebuttal has not changed my view of the weaknesses. In addition, I am ultimately still undecided on whether to recommend acceptance of this paper. As an empirical study, it does address the motivation set out in the paper itself. However, the findings are largely expected, and as other reviewers have also noted, the experimental scope is still limited. Although the authors have promised to add more experiments, I do not feel that this would make the work particularly surprising or compelling. Overall, I will maintain my current score.

**Key Questions For Authors:**

Given that the paper is well written and, in terms of the questions it raises and how thoroughly it addresses them, fairly complete, I do not have additional questions.

My only concern is that the privacy intuition being tested seems quite natural, the overall difficulty is lower than traditional model stealing, and therefore the results across the eight scenarios are reasonable and unsurprising.

The paper appears to be purely an empirical study, and I worry that it may be validating a point that would not surprise the community. For these reasons, I would give a weak accept, but with low confidence.

**Limitations:**

The paper discusses limitations in places, but not as a standalone section.

I suggest that, in future work (rather than in this revision), the authors improve the paper along two directions: (i) uncovering counter-intuitive findings; and (ii) methodological innovation.

**Strengths And Weaknesses:**

**Strengths:**
- It turns a privacy intuition into a testable study.
- The attack idea is consistent with model stealing and is technically sound.
- The presentation is good.

**Weaknesses:**
- Compared with traditional model stealing, the attack is predictably easier, and the underlying idea is not fundamentally new; thus the significance and originality are limited.
- The overall contribution is limited.

---

> ### Author Rebuttal · Authors · 2026-03-31
>
> **Weakness: Significance of Model Stealing in PMM**
>
> We thank the reviewer for highlighting the significance of our study and its relation to conventional model stealing attacks.
>
> We agree that our paper is primarily **an empirical study**, and we should have stated its contribution more explicitly in that context. Our main claim is not that PMM vulnerability is surprising in a binary sense, but that the field currently lacks a **systematic quantification of how much privacy leakage remains under partial sharing, which factors dominate it, and how the privacy-utility trade-off evolves as the number of shared layers changes**. This is the central scientific question of the paper.
>
> What is new is therefore not merely "showing that leakage exists”, but identifying a PMM-specific attack surface that is absent from standard full-sharing or single-task stealing settings. In PMM, different clients contribute models specialised to different tasks, only part of the model is shared, and the adversary seeks to reconstruct the victim’s **unshared private component** for a target task that may differ from its own. This is structurally different from conventional single-task model stealing and motivates our **8 prior-knowledge** scenarios.
>
> Another distinction is that conventional model stealing attacks primarily focus on **single-task** models without task heterogeneity. In contrast, our study is based on a setting in which different clients contribute models fine-tuned on **different tasks**. The adversary aims to reconstruct the victim’s model for a target task different from its own, thereby exploiting the information shared through partial model sharing. We focus on **cross-task knowledge extraction** rather than on conventional single-task model stealing. Therefore, our attack does not require access to the task-specific classification head, which is used only for evaluation.
>
> A notable finding of our study is that the adversary’s **cloned single-task model can outperform the merged multi-task model** on the target task in some cases. Although this may appear counterintuitive, it is consistent with the fact that the cloned model is task-specific, whereas the merged model must balance multiple tasks and therefore is subject to **task interference**. The finding suggests that privacy leakage can persist even when the merged model’s utility is not maximised, indicating that the leakage is mainly driven by partial model sharing and task specialisation rather than by merged-model utility alone. We also observe a strong performance of the clone model, even when an adversary has only 10\% of the victim’s data.
>
> A key difference from model stealing attacks in conventional collaborative learning is that PMM is a one-shot merging process: each client shares their model **only once** after local fine-tuning. As a result, the model stealing attack is more challenging in PMM since the adversary has limited prior knowledge and cannot access the victim’s training process or intermediate updates. Our attack is explicitly designed for this constrained PMM setting and outperforms conventional model stealing attacks when evaluating under the same attack scenarios and prior knowledge assumptions. This conclusion is supported by results from our experimental analysis in Table 3.
>
> Additionally, we would like to clarify that the main focus of this paper is to study the feasibility and effectiveness of model stealing in the context of **PMM**, which enables flexibility for clients to choose the number of layers to share. The number of shared layers is determined by PMM’s inherent trade-off between the global model utility contribution and the privacy leakage. To the best of our knowledge, this PMM setting has not been studied in prior work, which has primarily focussed on full model stealing in collaborative learning paradigms. The purpose of this work is to systematically investigate the empirical effectiveness of the model stealing attack in the context of PMM, and to provide PMM clients with actionable guidance for understanding the trade-offs from the adversary's perspective: keeping fewer layers private improves merged-model utility but also enlarges the attack surface; even limited leakage of data, model structure or parameters can materially strengthen cloning. We will revise the paper to present the contribution more clearly as a **first empirical characterization of the PMM privacy-utility frontier**, rather than as a purely methodological attack paper.

---

> > ### Author Rebuttal · Reviewer_dk1o · 2026-04-01
> >
> > Thank you to the authors for the time and effort invested in the rebuttal. After reading it carefully, my position remains the same as in my initial review. As I stated from the beginning, I did not have additional technical questions, but rather concerns about the paper’s overall contribution and impact. The rebuttal does not materially change this view, so I retain my original evaluation.

---

> > > ### Author Response · Authors · 2026-04-07
> > >
> > > We thank the reviewer for their feedback and will carefully incorporate their suggestions into the final version.

---

### Official Review · Reviewer_HdGW · 2026-03-08

**Soundness:** 3
**Presentation:** 3
**Significance:** 3
**Originality:** 3
**Overall Recommendation:** 5
**Confidence:** 3

**Summary:**

This work explores the privacy risks in model merging. Specifically, the authors formalize model clone attacks in the partial model merging scenario and propose a framework called ModelPirate to conduct such attacks. The authors perform experiments across eight attack scenarios to investigate the privacy risks arising from model merging.

**Compliance With Llm Reviewing Policy:**

Affirmed.

**Final Justification:**

The authors have addressed my concerns during the rebuttal process, and I will raise my score to 5. I encourage the authors to include more comprehensive experiments in the final version to better reflect the empirical nature emphasized in the title "An Empirical Study on the Resilience of Partial Merging to Model Clone Attacks."

**Key Questions For Authors:**

1.	Why do the experiments use only the default hyperparameters rather than tuning them adaptively for each setting?

2.	In Table 1, why is the adversary able to achieve higher accuracy than the client’s merged model in some scenarios?

**Limitations:**

yes

**Strengths And Weaknesses:**

The main advantages of the paper can be summarized as follows:

1.	This work focuses on the privacy of model merging, which is a new and interesting research topic.

2.	The paper provides a formalization of model clone attacks in the context of partial model merging and analyzes the threat model under different adversarial prior knowledge and objectives.

3.	The authors implement 8 different privacy attack scenarios and conduct experiments to demonstrate the potential privacy risks in model merging settings.

Despite its strengths, several issues remain, as outlined below:

1.	Since the paper is titled “An Empirical Study on the Resilience of Partial Merging to Model Clone Attacks,” the experimental evaluation should be more comprehensive. In particular, the authors could conduct experiments with a wider range of model architectures and different model merging methods. Although five datasets are mentioned, it appears that only two pairs of datasets are actually used in the experiments.

2.	The authors should repeat experiments multiple times and report the mean and variance of the results. Statistical significance tests (e.g., t-tests) would also help demonstrate the reliability of the reported findings.

---

> ### Author Rebuttal · Authors · 2026-03-30
>
> We would like to thank the Reviewer for constructive and thoughtful suggestions. Here are our responses to the concerns:
>
> **Model Architectures**
>
> We thank the reviewer for pointing out the limited range of model architectures in our experiments. To address this concern, we have performed a supplementary experiment with LLaMA-1B using the same datasets as in the experiments on the T5 model in Appendix C. On the IMDB dataset, the clone model accuracy achieves **94.90%** in the unknown dataset scenario and **95.58%** in the partially known dataset setting. The results are close to the official reported accuracy of the LLaMA-1B model fine-tuned on the IMDB dataset, which is *96.04%* as shown in [1]. This demonstrates that our attack remains effective beyond the model architectures used in the main paper and is also effective on LLMs. We will include detailed experimental setups and results, along with a discussion, in the final version of this paper.
>
> **Model Merging Methods**
>
> We agree with the reviewer that more advanced model merging methods (TIES[2], DARE[3]) may improve the merged model performance. In response, we have conducted supplementary experiments comparing multiple representative merging methods. The results of the merged model accuracy are summarised in the table below:
>
> ||ViT-B32|ViT-B16|ViT-L14|
> |-|-|-|-|
> |TA|79.68\%|81.55\%|89.63\%|
> |TIES|63.02\%|67.59\%|77.29\%|
> |DARE|77.19\%|79.40\%|87.40\%|
>
> These results show that the choice of merging method can affect the performance of the merged model. In particular, **Task Arithmetic (TA)** achieves the best performance among all methods evaluated.
>
> However, the primary objective of this work is **not to optimise the merged model performance**, but to study the privacy risks (quantified by **cloned model performance**) associated with sharing different portions of the model parameters in PMM. We would like to emphasise that our attack is **compatible with any partial model merging scheme** that requires clients to share part of their model parameters, such as TA, TIES, and DARE. Therefore, our primary goal is to evaluate the privacy vulnerability introduced by the act of partial parameter sharing, independent of the specific merging method used. We will revise the paper to clarify this scope.
>
> **Datasets and Random Seeds**
>
> To further demonstrate the **generality** of our study, we have also expanded our main experiment by using **all five datasets** considered in model merging as target datasets. In addition, to assess the robustness and reliability of the results, we repeated each experiment using **ten random seeds** and report the mean and standard deviation in the table below:
>
> ||MNIST|DTD|EuroSAT|GTSRB|SVHN|
> |-|-|-|-|-|-|
> |Mean|99.65%|93.20%|99.69%|98.36%|97.42%|
> |Std|0.016%|0.43%|0.063%|0.075%|0.035%|
>
> The results above show that the cloned model accuracy is consistently high across ten random seeds.
>
> ---
>
> **Q1: Hyperparameters Used**
>
> We thank the reviewer for raising the important point about hyperparameter selection. The hyperparameters in our experiments were carefully selected through preliminary tuning to ensure that the clone model reliably converges within 1500 training rounds. We use 10% of the victim data as the adversary’s known data by default to simulate an attack setting in which the adversary has only limited access to the victim's data. To ensure a fair comparison, we **vary one hyperparameter at a time** while keeping the others unchanged. We also record intermediate results at 500 and 1000 rounds to ensure the model's stability under various settings.
>
> ---
>
> **Q2: Clone Accuracy Higher than Merged Accuracy**
>
> We appreciate the opportunity to clarify this point. Merged accuracy and clone accuracy measure the performance of two fundamentally different models:
>
> - *Merged accuracy* is the accuracy of the victim’s merged **multi-task** model on the target task. Since the merged model combines capabilities from multiple task-specific models, its performance on any individual task can be reduced relative to the corresponding task-specific fine-tuned model.
>
> - *Clone accuracy* refers to the accuracy of the adversary’s cloned model on the target task. The cloned model is a **single-task** model dedicated to a specific target task and therefore does not suffer from the cross-task interference inherent in multi-task merging. As a result, the clone accuracy can exceed the merged accuracy in some cases.
>
> Therefore, it is not contradictory for the adversary’s **clone accuracy** to be higher than the victim’s **merged accuracy** on a specific task because the merged model combines the capability of multiple single-task models, whereas the cloned model is dedicated to a specific task.
>
> [1]. huggingface.co/yash3056/Llama-3.2-1B-imdb
>
> [2]. Yadav et al. (NeurIPS 2023). TIES-Merging: Resolving Interference When Merging Models
>
> [3]. Yu et al. (ICML 2024). Language models are super mario: absorbing abilities from homologous models as a free lunch

---

> > ### Author Rebuttal · Reviewer_HdGW · 2026-04-02
> >
> > Thank you for the detailed response and the additional experiments. The authors have addressed my concerns.

---

> > > ### Author Response · Authors · 2026-04-03
> > >
> > > We thank the reviewer for the positive feedback and for increasing the score. We are pleased that the clarifications were helpful, and we will carefully incorporate the reviewer’s suggestions in the final version.

---

### Official Review · Reviewer_XacF · 2026-03-13

**Soundness:** 3
**Presentation:** 3
**Significance:** 2
**Originality:** 3
**Overall Recommendation:** 5
**Confidence:** 3

**Summary:**

The work demonstrates the privacy vulnerabilities of partial model merging (PMM) where front layers of several models are merged to improve the model generalization and rear layers are kept private to ensure the model IP, yet the functionality of rear layers can be easily reconstructed with the access to the private model service. The proposed model clone attack, namely ModelPirate, is shown effective on private model clone, given varying degrees of prior knowledge. Evaluations are conducted on vision and language tasks over light but hot model architectures, such as ViT-B and T5 models. Results demonstrate the effectiveness of the proposed model clone attack, especially when the knowledge of partial datasets on the victim model is acquired by the attackers.

**Compliance With Llm Reviewing Policy:**

Affirmed.

**Final Justification:**

I decided to raise my score to 5. The authors well addressed my concerns. PMM suffers from the accuracy degradation by increasing the generalization; while this tradeoff could make the PMM not a perfect solution to all scenes (gain without loss), it should find scenarios to be applied. In which case, the privacy analysis in this work presents a meaningful guidance.

**Key Questions For Authors:**

Please see the weaknesses.

Other comments:
1. I noticed in Figure 10, interestingly, the clone accuracy dropped at the last several layers, which shows different trend from other figures. Could the authors give some analysis on this phenomenon?
2. In default PMM procedure, after the client collected the merged part, why the client doesn't further fine-tune the model on the specific task, so that the accuracy can recover and generalization can be reserved?

**Limitations:**

yes

**Strengths And Weaknesses:**

**strengths**

1. The PMM topic is emerging in recent years, thus its privacy issue is open to investigate. This work tries to fill this gap by evaluating model clone on PMM setup with different knowledge one attacker holds with.
2. The proposed attack mechanism is reasonable and easy to follow,. Even I don't expertise in the PMM topic I could follow the background and attack flow.
3. The evaluated tasks on vision and NLP are practical and widely applied currently. Although the evaluated models are still shallow due to expensive training cost on large models, the results are still instructive.

**weaknesses**

1. While not from the proposed attack itself, I have concern of the practice of PMM. Since the observed great accuracy degradation of model merging (like over 30% at most) compared to direct fine-tuning, the clue of how PMM should be deployed in practice is still not clear. In current model development, even marginal accuracy increment is a big deal, while PMM could easily degrade the good performance.
2. In the proposed attack mechanism, the victim model output, i.e., the logits/softmax results, should be revealed to the attacker, for the sake of MAE computation. This condition is not easy to achieve under a black-box setting, where only the category index (argmax) is returned in real service.

---

> ### Author Rebuttal · Authors · 2026-03-30
>
> We would like to thank the Reviewer for their careful review of our manuscript. Here are our responses to the concerns:
>
> ---
>
> **Weakness1: Practice of PMM**
>
> We appreciate the reviewer's concern regarding the accuracy degradation of PMM. We would like to clarify that PMM constructs a single **multi-task** merged model by partially merging multiple **single-task** fine-tuned models on different tasks. The goal of PMM is to obtain a unified model that combines the capabilities of all original fine-tuned models, so that only one model needs to be stored and deployed for inference, without requiring the exchange of domain-specific data or task-specific models.
>
> The observed performance drop relative to the original fine-tuned models is mainly due to **task interference**, in which model parameters optimised for different tasks may conflict during merging. Therefore, model utility degradation on each individual task is expected after merging. Importantly, the 30\% performance degradation corresponds to an extreme **worst-case setting** with the minimum number of layers merged. In practice, when a larger fraction of layers is merged to balance privacy and utility, the utility degradation is substantially smaller. For example, under our default configuration, where 75\% of layers are merged, the **utility drop is less than 10\%** relative to FMM. Note that fewer layers need to be merged when task interference is lower. However, a detailed task-level analysis is beyond the scope of the work.
>
> ---
>
> **Weakness2: Attack Practicability**
>
> We thank the reviewer for the thoughtful comment. We would like to clarify that our proposed attack is **specifically designed for PMM**, where the merged model itself **does not include the classification head**. In practical use, a **task-specific** local classification head is attached to the merged model during inference for the task the user intends to perform. Therefore, the black-box victim model output that the adversary aims to clone is the output of the last layer before the classification head. We describe the clone model training procedure in Sec.3.3, with footnote 6 explaining that the classification head for the victim's task $T_v$ is used **only for performance evaluation** (i.e., MAE computation) and not required during clone model training. We will further clarify this setting in the final version to avoid confusion.
>
> ---
>
> **Question1: Trend in Figure 10**
>
> The T5 model used to produce Figure 10 consists of an encoder and a decoder, and these two components exhibit different trends in clone accuracy. The trend in the encoder’s clone accuracy is similar to that observed for the ViT models. The reason is that the encoder in the T5 model progressively compresses information from the input into task-specific representations in later layers, whereas the decoder expands and transforms information already extracted by the encoder. The information bottleneck lies between the encoder and the decoder, such that no additional information can be added to the encoder's outputs. This makes the decoder component relatively easier to clone and results in consistently higher clone accuracy than the encoder.
>
> Comparatively, ViT is an **encoder-only model** with the classification head directly attached to the encoder output. As in the T5 encoder, layers closer to the outputs extract more task-specific features. Therefore, the clone accuracy for the ViT models is expected to follow the same trend as the T5 encoder, and the results confirm this expectation. We will provide a more detailed analysis in the final version to explain the reason behind this phenomenon.
>
> ---
>
> **Question2: Merging Procedure**
>
> In PMM, each client fine-tunes a common pre-trained model on its own task using the local data, producing a set of **task-specific models**. These models are optimised for a specific task and therefore generalise poorly outside their individual tasks. Model merging is a **training-free** approach that combines the capabilities of multiple single-task models into a single multi-task model without additional fine-tuning. Therefore, the purpose of PMM is to obtain a unified model with **stronger cross-task generalisation** and inherit capabilities from all single-task models without incurring additional training costs.
>
> We do not claim that PMM aims to outperform any original single-task model. However, although the merged model's utility on a specific task is lower than that of the corresponding single-task model, it performs better across other tasks. This claim has been supported empirically in Figs. 2 and 9, as well as in Table 2 of the paper.
>
> We agree that the post-merging fine-tuning can improve task-specific performance. But this would **no longer be training-free**. More importantly, such fine-tuning is **at the expense of reducing cross-task generalisation**, as it would likely bias the model towards the specific task and increase task interference with respect to other tasks.

---

> > ### Author Rebuttal · Reviewer_XacF · 2026-04-02
> >
> > The authors well addressed my concerns, thus I decided to raise my score as my final recommendation. I suggest the authors give more comprehensive introduction on PMM to emphasize its scenarios in practice, in their revision.

---

> > > ### Author Response · Authors · 2026-04-03
> > >
> > > We thank the reviewer for the positive feedback and for raising the score. We are glad that the clarifications were helpful, and we will carefully incorporate the suggestions into the final version.

---

### Decision · Program_Chairs · 2026-04-30

**Decision:**

Accept (regular)

**Comment:**

With systematic experiments, the authors show that the comprehensive experiments reveal that merging NNs without adequate protection is highly vulnerable. Even when only a small fraction of training data, model parameters, or model structure is exposed, adversaries can still recover significant portions of the private model's performance. Reviewers agree that the experiments are extensive and up to date. It is true, as pointed out by one of the reviewers, that partial merging is not particularly designed for privacy or security. However, perhaps the paper has enough contributions, despite weaker motivations than the paper claims.